# A method for comparing MRI sequences of the knee for segmentation based on morphological features

**Yunsub Jung**[1,2], **Morten Bilde Simonsen**[1,2], **Michael Skipper Andersen**[1,2]*

**1** Department of Materials and Production, Aalborg University, Aalborg, Denmark, **2** Department of Materials and Production, Center for Mathematical Modeling of Knee Osteoarthritis, Aalborg University, Aalborg, Denmark

* msa@mp.aau.dk

**Data Availability Statement:** Data cannot be shared publicly because of contain potentially sensitive information for patients. Data are available from the Danish Ethics Committee's

## Abstract

### Background

In magnetic resonance imaging (MRI) segmentation research, the choice of sequence influences the segmentation accuracy. This study introduces a method to compare sequences. By aligning sequences with specific segmentation objectives, we provide an example of a comparative analysis of various sequences for knee images.

### Methods

Based on the profile information of virtual rays, we devised metrics to compute the edge sharpness and contrast. Edge analysis was performed in five edges ($E_{BB}$: between cancellous and cortical bone, $E_{BC}$: between cortical bone and cartilage, $E_{CF}$: between cartilage and fat, $E_{CM}$: between cartilage and meniscus, $E_{BT}$: between cortical bone and tissue). Subsequently, profiles were extracted from the virtual ray that traversed the defined edge. Finally, edge characteristics were compared in each sequence using the computed metrics.

### Results

In the case of sharpness, T1-weighted (T1) showed the highest at $E_{BB}$, $E_{CF}$, and $E_{BT}$ (all, $p < .05$). The fat-suppressed 3D spoiled gradient-echo (SPGR) was the highest at $E_{BC}$, and proton density fat-saturated (PDFS) was the highest at $E_{CM}$ (all, $p < .005$). Depending on each sequence, the knee structures showed different edge characteristics. Also, it was confirmed that the edge properties of the structure depend on the adjacent materials.

### Conclusions

The ultimate goal of this study is to present a methodology for selecting the most appropriate MRI sequence for segmentation, which can be applied to images of other parts in addition to the knee images used in the study. The method we present quantitatively evaluates the edge characteristics, and experimental results show that our method shows consistent

(Scientific Ethics Committee for the Region Nordjylland, contact via vek@rn.dk) for researchers who meet the criteria for access to confidential data.

**Funding:** This work supported by the Novo Nordisk Foundation (grant no. NNF21OC0065373). The funders had no role in study design, data collection and analysis, decision to publish, or preparation of the manuscript.

results according to the edge. Our method will provide additional information for MRI sequence selection for segmentation.

---

## 1. Background

Segmentation of various structures of the human body from medical imaging has applications in numerous fields other than diagnosis and quantification of lesions, including radiation treatment planning [1], 3D printing of human structures [2], and extracting anatomical geometries for biomechanical modeling [3]. While manual segmentation can extract anatomical geometries, its practical utility is limited by its time-consuming nature. Therefore, semi-automatic and fully automatic algorithms have been developed to extract various human body structures.

Image quality is an important factor in determining automatic segmentation performance [4], where even the same segmentation algorithm has varying performance depending on the image quality [5, 6]. In general, image quality is determined by the hardware system of a medical device, scan parameters, various noises, and external environmental variables such as movement artifacts and implant substances in the human body [7]. It is difficult to change actions regarding hardware systems and external environment variables. Still, scan parameters are one of the variables that can easily be adjusted to the image quality and can be changed according to the purpose of scanning. Since computed tomography (CT) and magnetic resonance imaging (MRI) are the technologies that best show the human body's anatomical information, most automatic segmentation studies mainly use these two technologies. CT image quality depends on radiation dose, which affects the clarity of the materials in the image [8]. However, since the dose is typically within a specific range for each body region, it does not significantly affect the imaging quality of specific materials in the body. For MRI, on the other hand, the sequence is crucial for image quality. Even with the same material, depending on the MRI sequence, the quality of the image changes dramatically [9].

MRI of the knee can be used to evaluate changes in knee structures for diseases such as knee osteoarthritis (KOA) and is preferred over CT because MRI provides better soft tissue contrast than CT [10]. To evaluate KOA progression, observing the cartilage, meniscus, and soft tissue associated with KOA is important, and this observation includes visualization and quantitative measurements of structures [10]. Various studies have focused on segmenting knee structures, from existing analytic-based methods to recent machine learning-based methods [11, 12].

Selecting the sequence is important in MRI image structure segmentation studies [11, 13–17]. Existing knee structure segmentation studies have been conducted using diverse sequences [11, 13]. However, these studies mentioned the MRI sequence for the data used, but none stated the reason for choosing the MRI sequence used. Existing studies include visual evaluation of knee structures according to sequence [14–16] and evaluation of segmentation performance by sequence using the same segmentation algorithm [6, 17]. However, although visual evaluation can be a basic guide for selecting sequences, it does not provide quantitative values for finding the best-suited sequences with the best segmentation performance. Typically, segmentation algorithms are developed and tuned using a specific sequence. Therefore, it may not be a good idea to use the performance of automatic segmentation to determine the best sequence. This is because the sequence used in the development phase will likely show high performance.

The edge sharpness and contrast between two tissues are the most important image qualities determining segmentation performance [18]. These are essential image features in analytic-based methods designed by human engineers and machine learning-based methods in which optimal features are automatically extracted and computed. This study introduces a novel metric to compare MRI sequences to identify the most suitable for segmenting human body structures by quantitatively analyzing edge sharpness and contrast. Different MRI sequences of the knee will be compared to demonstrate our metrics. The example is not intended to identify the optimal knee sequence but instead serves for demonstration purposes. This study aims to serve as a resource aiding in selecting appropriate MRI sequences for segmentation algorithms.

## 2. Methods

### 2.1. Data acquisition

The study was conducted in accordance with the ethical regulations of the North Denmark Region Committee on Heath Research Ethics, Denmark. Also, the experimental protocol was explained to all participants, and written informed consent was obtained. MRI 3T (Signa HDxt 3.0T, GE Healthcare, USA) scans were performed on 11 healthy participants without knee disease or surgery history. Healthy participants were chosen, given that damaged structures, such as bone and cartilage, may deviate from the distinctive characteristics of the structures. The participants were, on average, 40.6±16.3 years old, weighed 74.8±11.8 kg, and were 178.0±7.3 cm tall. The same location on the knee, including the femur and tibia, was scanned using three different MRI sequences (Fig 1). In order to obtain the same anatomical structure image in the three sequence images, preliminary practice was conducted to prevent positional changes due to the participants' movement, and then the scan was performed. The three sequences used in this study are T1-weighted (T1), proton density fat-saturated (PDFS), and fat-suppressed 3D spoiled gradient-echo (SPGR) (Table 1). T1 sequence mainly evaluates bone marrow and bone tumors because they contrast perfectly for cortical, marrow, and surrounding tissues [19]. The PDFS sequence has good contrast for the cartilage surface, so it is useful for measuring cartilage thickness or examining cartilage abnormalities [20]. Finally, the SPGR is known to have good contrast between cartilage and surrounding bone through fat suppression and has a high contrast-to-noise ratio [21]. Generally, since the segmentation task usually

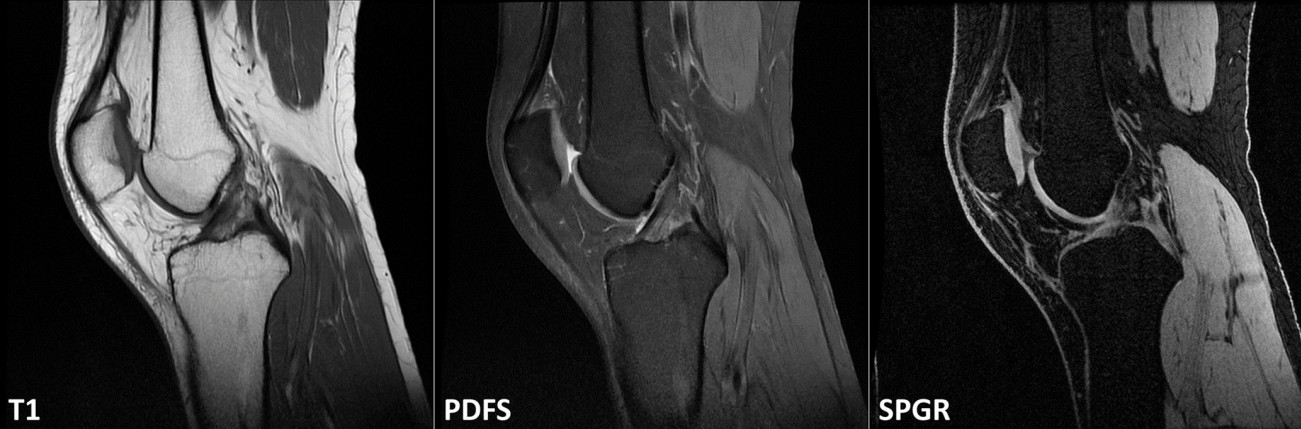

**Fig 1. MRI images of the three sequences used for evaluation.** All sequence images were scanned to include the same knee structure and reconstructed in the sagittal plane. (T1: T1-weighted, PDFS: proton density fat-saturated, SPGR: fat-suppressed 3D spoiled gradient-echo).

**Table 1. Scan protocol and image description.**

| Sequence | TR (msec) | TE (msec) | ETL | FA (°) | Image | | | |
|---|---|---|---|---|---|---|---|---|
| | | | | | Matrix (pixel) | Slice thickness (mm) | Slice spacing (mm) | Pixel size (mm) |
| T1 | 818 | 12.5 | 4 | 90 | $512 \times 512$ | 3 | 3.5 | $0.3125 \times 0.3125$ |
| PDFS* | 3928 | 21.16 | 8 | 110 | $512 \times 512$ | 3 | 3.5 | $0.3125 \times 0.3125$ |
| SPGR† | 9.884 | 4.16 | 1 | 25 | $512 \times 512$ | 1 | 0.5 | $0.3125 \times 0.3125$ |

Note.-The images used for analysis were reconstructed in the sagittal plane. TR = repetition time, TE = echo time, ETL = echo train length, FA = flip angle, T1 = T1-weighted, PDFS = proton density fat-saturated, SPGR = fat-suppressed 3D spoiled gradient-echo.

uses already acquired images for clinical purposes, we did not separately set MRI scan parameters for this study. We used MRI scan parameters set for clinical purposes in the hospital without any changes. For quantitative analysis, images with the same anatomical location were manually selected from three sequences, and 12 images per participant (4 images from each sequence) were extracted. The four selected images consisted of two that showed bone structures well and two that showed cartilage structures.

## 2.2. Manual drawing of edge

The first step of the quantitative edge analysis was extracting the edges of the bone and cartilage. In the three sequence images, the edges of cortical bone, cancellous bone, and cartilage were manually drawn, respectively (Fig 2). In order to classify the edges according to material, manually drawn edges were defined according to the materials on both sides of the edge ($E_{BB}$: the edge between the cancellous bone and cortical bone, $E_{BC}$: the edge between cortical bone and cartilage, $E_{CF}$: the edge between cartilage and fat, $E_{CM}$: the edge between cartilage and meniscus, $E_{BT}$: the edge between cortical bone and tissue, Fig 3). In $E_{BT}$, the tissue mainly contains fat or muscle. Tibia's $E_{CF}$ was not included because there was no fat material or only a small area in contact with the tibia cartilage. The manually drawn edges become the target edge for evaluating the characteristics of edges according to the sequence. For this purpose, the manual edges were drawn in the three sequence images.

## 2.3. Quantitative analysis of edge using virtual ray

The overall process for sequence evaluation at edges defined by manual marking consists of the following steps. (1) Computing of the unit normal vectors, (2) Virtual ray propagation, (3) Extraction of pixel profile from virtual ray, (4) Profile interpolation, (5) Computing a sharpness and contrast.

Unit normal vectors ($\hat{n}$) were calculated [22] in the orthogonal direction from all edge points of the cancellous bone obtained from the manual drawings (Eq 1, Fig 2(c)).

$$\hat{n} = \frac{1}{\sqrt{dx^2 + dy^2}} \begin{bmatrix} -dy \\ dx \end{bmatrix} \tag{1}$$

Here, $dx$ and $dy$ refer to derivatives in the x and y directions of the image pixels, respectively. At all edge points of the cancellous bone, virtual rays ($\vec{l} = \hat{n}\alpha$) with a length of 10 mm or more ($|\alpha| > 10$ mm) were created in the direction of the computed unit normal vector. The points where the virtual ray and the manual edges meet become $E_{BB}$, $E_{BC}$, $E_{CF}$, $E_{CM}$, and $E_{BT}$, respectively (Fig 3). This virtual ray terminates when a depth of 2 mm is reached in the inside bone of cancellous bone fat, meniscus, and tissue regions. Each virtual ray stores the pixel

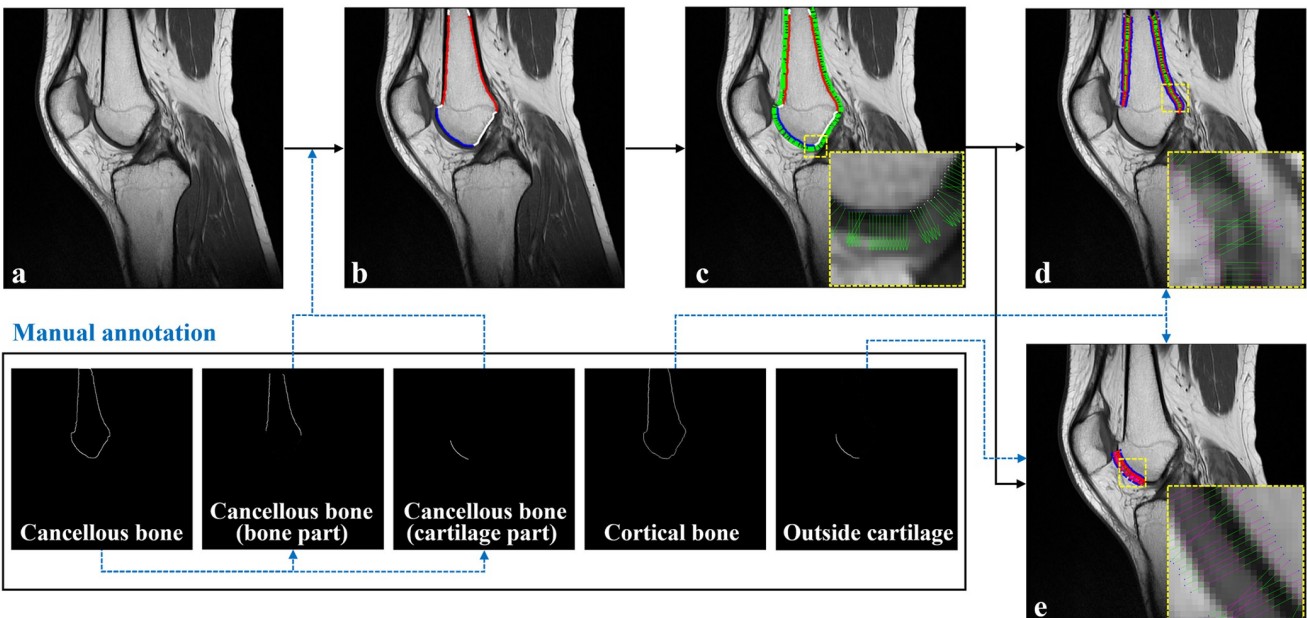

**Fig 2. The analysis process of edge characteristics.** Edges of cancellous bone, cortical bone, and cartilage are obtained by manual marking. (a) Original T1-weighted image, (b) The edge of cancellous bone is divided into bone and cartilage regions, respectively, and (c) the virtual ray generated from the cancellous bone is propagated to both sides in the orthogonal direction. The point where this virtual ray meets the (d) cortical bone and (e) cartilage obtained by manual marking is calculated, and the virtual ray ends at a certain depth of fat, muscle, and meniscus.

value of the image through which the ray passes, and a one-pixel value is extracted from one image pixel (Fig 4). This profile curve with pixel information is interpolated by the cubic spline method [23] with a point interval of 1/10 of a pixel. The sharpness ($S_j$) and contrast of the edges are computed through analysis of this profile. First, the sharpness value of $E_{XX}$ (i.e., $E_{BB}$, $E_{BC}$, $E_{CF}$, $E_{CM}$, and $E_{BT}$) becomes the differential coefficient value at the point of $E_{XX}$. Then, the contrast value of $E_{XX}$ is computed as the ratio of the average pixel values in both materials ($S_1$ and $S_2$) in $E_{XX}$ (Eq 2), where $S_j$ represents the average pixel values of a single material (Eq 3).

$$\text{Contrast} = \begin{cases} \dfrac{S_1}{S_2}, & \text{if } (S_1 > S_2) \\ \dfrac{S_2}{S_1}, & \text{if } (S_2 > S_1) \end{cases} \tag{2}$$

$$S_j = \sum_{i=0}^{n} P_i / \Delta l \tag{3}$$

Here, $P_i$ represents the pixel value of a material, $\Delta l$ means the thickness of the material appearing in the image, and $n$ is the number of interpolated pixels in the $j$th material.

### 2.4. Inter-subject variability

The inter-subject variability (*ISV*) was calculated to determine the difference in measured values among the participants. The average (*ISV$_{AVE}$*) and maximum (*ISV$_{MAX}$*) of *ISV* were

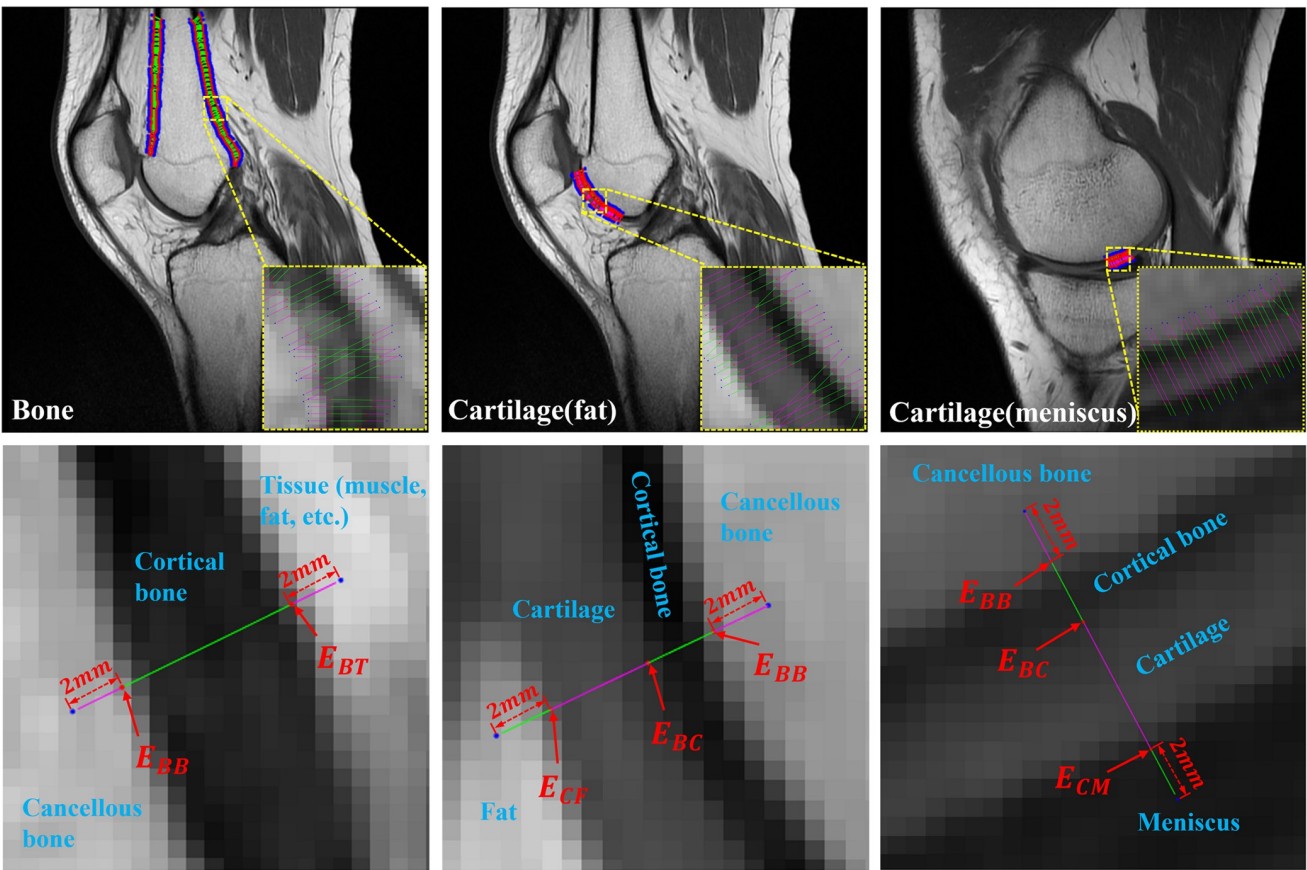

**Fig 3. Analysis of the edge characteristics according to the materials on both sides of the edge.** ($E_{BB}$: the edge between cancellous bone and cortical bone, $E_{BC}$: the edge between cortical bone and cartilage, $E_{CF}$: the edge between cartilage and fat, $E_{CM}$: the edge between cartilage and meniscus, $E_{BT}$: the edge between cortical bone and tissue).

calculated as Eqs 4 and 5. The average ($S_{AVE}$ or $C_{AVE}$) and standard deviation ($S_{SD}$ or $C_{SD}$) of sharpness (or contrast) were obtained from 11 participants. The $\sigma_{MAX}$ is the largest difference from the mean among the 11 measured values.

$$ISV_{AVE} = \left( \frac{S_{SD}}{S_{AVE}} \right) \cdot 100 \qquad (4)$$

$$ISV_{MAX} = \left( \frac{|\sigma_{MAX} - S_{AVE}|}{S_{AVE}} \right) \cdot 100 \qquad (5)$$

## 2.5. Data analysis

The quantitative analysis method of edge, including the image processing algorithm proposed in this study, was implemented using a software development tool (MATLAB, 2019a, Math-Works, USA) and an image analysis software (ImageJ, 1.53k, National Institutes of Health, USA) was used for manual drawing. The normality was tested using Kolmogorov-Smirnov (p > .05). One-way repeated measures ANOVA was used to compare the three MRI sequences at

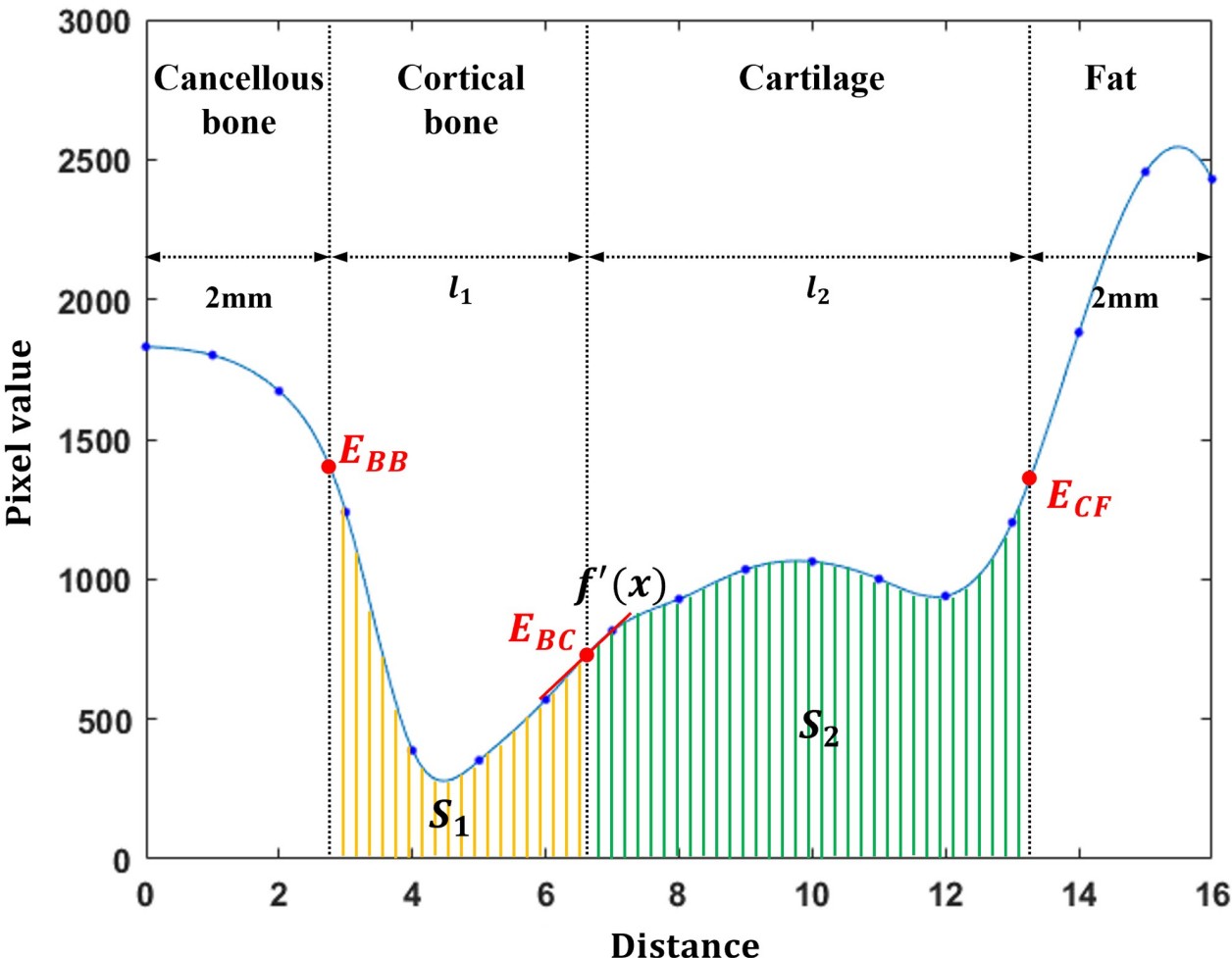

**Fig 4. This shows the profile by a virtual ray crossing the cancellous bone, cortical bone, cartilage, the fat regions.** The virtual ray contains pixel information up to 2mm outside of $E_{BB}$ and $E_{CF}$. The blue dot represents a pixel value obtained from each image pixel, and the blue line is the interpolated curve by cubic spline interpolation. Sharpness and contrast are calculated at all edges. ($E_{BB}$: the edge between cancellous bone and cortical bone, $E_{BC}$: the edge between cortical bone and cartilage, $E_{CF}$: the edge between cartilage and fat, $E_{CM}$: the edge between cartilage and meniscus, $E_{BT}$: the edge between cortical bone and tissue).

each edge ($E_{BB}$, $E_{BC}$, $E_{CF}$, $E_{CM}$, and $E_{BT}$). In addition, post hoc comparisons were performed with Bonferroni correction. All statistical analyses were performed using statistical software (SPSS, version 28.0.1.1, IBM Corp., USA), and a P-value of less than .05 was considered statistically significant.

## 3. Results

The edge sharpness and contrast results computed by the proposed metrics are presented in Fig 5, Tables 2 and 3 (S1 and S2 Tables). The sharpness and contrast results in $E_{BB}$, $E_{BC}$, $E_{CM}$, and $E_{BT}$ showed similar patterns for the femur and tibia regions. In the case of sharpness, T1 was the highest at the $E_{BB}$ edge (all, $p < .001$), SPGR was the highest at the $E_{BC}$ edge, T1 was the highest at the $E_{CF}$ edge, PDFS was the highest at the $E_{CM}$ edge ($p < .005$ at the femur, $p < .001$ at tibia), and T1 was the highest at the $E_{BT}$ edge ($p < .01$ at the femur, $p < .05$ at tibia). In the case of contrast, T1 was the highest at the $E_{BB}$ edge (all, $p < .05$). At the $E_{BC}$ edge, PDFS

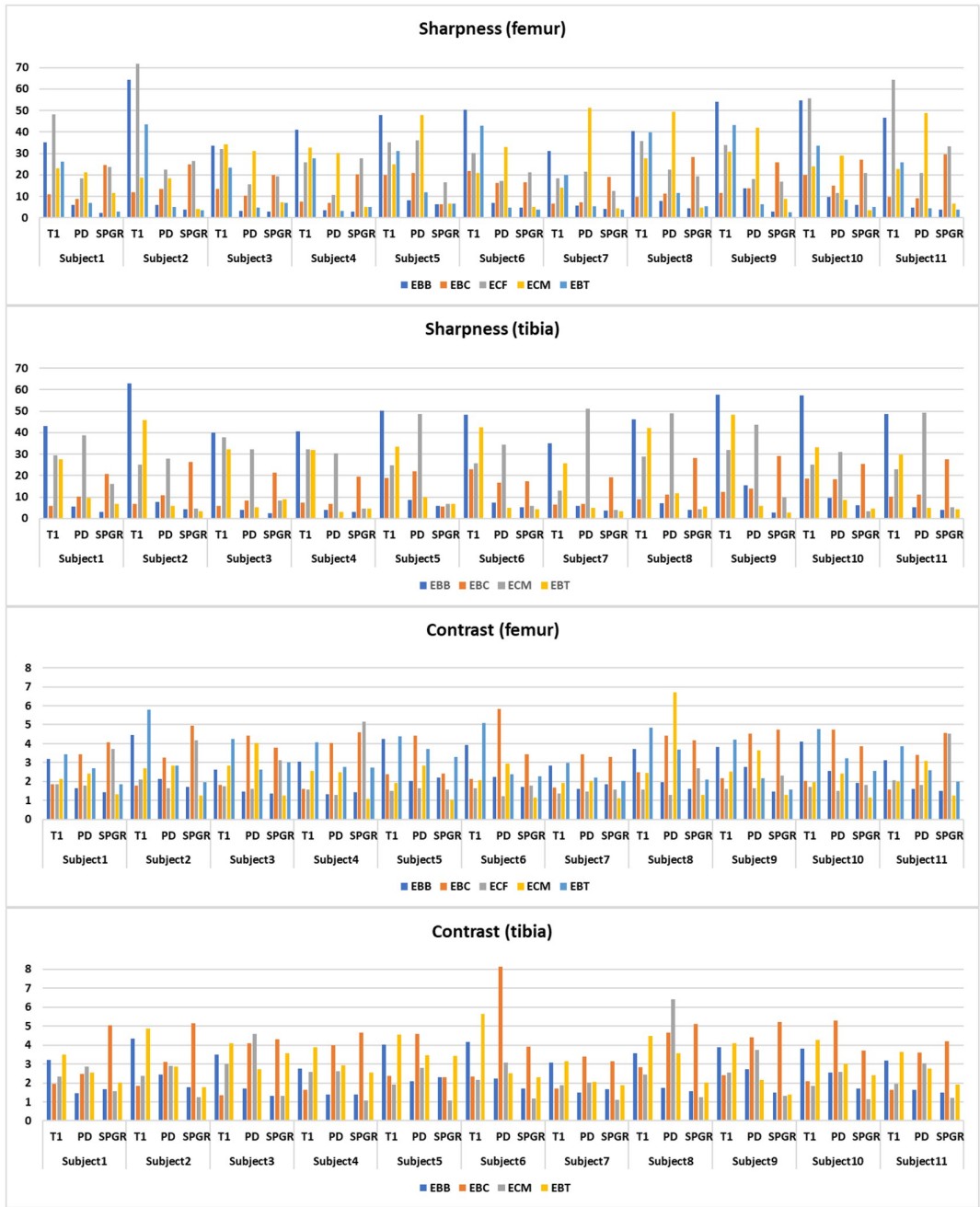

**Fig 5. Results of sharpness and contrast were obtained from 11 participants.** (T1: T1-weighted, PDFS: proton density fat-saturated, SPGR: fat-suppressed 3D spoiled gradient-echo, $E_{BB}$: the edge between cancellous bone and cortical bone, $E_{BC}$: the edge between cortical bone and cartilage, $E_{CF}$: the edge between cartilage and fat, $E_{CM}$: the edge between cartilage and meniscus, $E_{BT}$: the edge between cortical bone and tissue).

and SPGR showed similar levels, and these two values were higher than T1 (PDFS $\approx$ SPGR > T1). SPGR was the highest at the $E_{CF}$ edge (all, p < .005), PDFS was the highest at the $E_{CM}$ edge (all, p < .005), T1 was the highest at the $E_{BT}$ edge (p < .001 at the femur, p < .005 at tibia).

**Table 2. Results of the edge sharpness and contrast.**

| | Sequence | Femur | | | | | Tibia | | | |
|---|---|---|---|---|---|---|---|---|---|---|
| | | $E_{BB}$ | $E_{BC}$ | $E_{CF}$ | $E_{CM}$ | $E_{BT}$ | $E_{BB}$ | $E_{BC}$ | $E_{CM}$ | $E_{BT}$ |
| Sharpness | T1 | 45.36±10.22 | 13.01±5.28 | 41.00±16.73 | 24.80±6.05 | 32.49±8.65 | 48.19±8.50 | 11.27±6.10 | 26.98±6.35 | 35.66±7.70 |
| | PDFS | 6.91±2.97 | 12.12±4.27 | 19.48±6.77 | 36.53±11.78 | 6.59±2.91 | 7.32±3.24 | 12.32±4.85 | 39.65±8.91 | 6.75±2.67 |
| | SPGR | 4.03±1.29 | 22.06±6.65 | 21.64±5.90 | 6.13±2.39 | 4.54±1.40 | 4.05±1.24 | 21.81±6.78 | 6.64±3.69 | 5.04±1.82 |
| Contrast | T1 | 3.55±0.61 | 1.95±0.31 | 1.71±0.23 | 2.28±0.34 | 4.33±0.78 | 3.59±0.50 | 2.01±0.44 | 2.27±0.36 | 4.19±0.68 |
| | PDFS | 1.94±0.46 | 4.17±0.77 | 1.53±0.20 | 3.22±1.29 | 2.81±0.53 | 1.94±0.47 | 4.34±1.49 | 3.33±1.22 | 2.77±0.47 |
| | SPGR | 1.65±0.26 | 3.99±0.74 | 2.95±1.27 | 1.21±0.10 | 2.30±0.53 | 1.64±0.26 | 4.24±0.94 | 1.22±0.14 | 2.29±0.67 |

Note.-Results represent mean ± standard deviation, T1 = T1-weighted, PDFS = proton density fat-saturated, SPGR = fat-suppressed 3D spoiled gradient-echo, $E_{BB}$: edge between cancellous bone and cortical bone, $E_{BC}$: edge between cortical bone and cartilage, $E_{CF}$: edge between cartilage and fat, $E_{CM}$: edge between cartilage and meniscus, $E_{BT}$: edge between cortical bone and tissue

The results for $ISV_{AVE}$ and $ISV_{MAX}$ are presented in Table 4. There was no significant difference in $ISV_{AVE}$ values according to the measurement location (femur or tibia), but there was a difference according to the measurement index. In the case of sharpness, the average of the $ISV_{AVE}$ values of T1, PDFS, and SPGR is 30.20, 27.23, and 34.74, respectively, showing a larger contrast (T1: 16.32, PDFS: 25.15, SPGR: 20.76). For $ISV_{MAX}$, no pattern in measurement location or metric was observed, but some participants' measurement values showed a large difference.

## 4. Discussion

The MRI sequence is one of the important parameters that determine image characteristics. Many clinical studies have been conducted to find the sequence most suitable for diagnostic purposes [14–16]. In addition, studies on auto-segmentation algorithms have been conducted using various sequences [11, 12]. However, the most suitable MRI sequence for effective structure segmentation has not been considered. The present study presents a new method to find the most suitable MRI sequence for human body structure segmentation through quantitative edge sharpness and contrast analysis.

The proposed metrics confirmed that the T1 shows good sharpness and contrast in the edge between cortical bone and cancellous bone and between cortical bone and tissue (fat,

**Table 3. P-value of the edge sharpness and contrast among the sequences.**

| | Pairwise* | Femur | | | | | Tibia | | | |
|---|---|---|---|---|---|---|---|---|---|---|
| | | $E_{BB}$ | $E_{BC}$ | $E_{CF}$ | $E_{CM}$ | $E_{BT}$ | $E_{BB}$ | $E_{BC}$ | $E_{CM}$ | $E_{BT}$ |
| Sharpness | T1-PDFS | $< .001^*$ | 0.766 | $< .001^*$ | $0.002^*$ | $< .001^*$ | $< .001^*$ | 0.382 | $< .001^*$ | $< .001^*$ |
| | T1-SPGR | $< .001^*$ | $0.002^*$ | $< .001^*$ | $< .001^*$ | $< .001^*$ | $< .001^*$ | $< .001^*$ | $< .001^*$ | $< .001^*$ |
| | PDFS-SPGR | $< .001^*$ | $< .001^*$ | 0.989 | $< .001^*$ | $0.007^*$ | $< .001^*$ | $< .001^*$ | $< .001^*$ | $0.028^*$ |
| Contrast | T1-PDFS | $< .001^*$ | $< .001^*$ | $0.005^*$ | $0.003^*$ | $< .001^*$ | $< .001^*$ | $< .001^*$ | $< .001^*$ | $< .001^*$ |
| | T1-SPGR | $< .001^*$ | $< .001^*$ | $< .001^*$ | $< .001^*$ | $< .001^*$ | $< .001^*$ | $< .001^*$ | $< .001^*$ | $< .001^*$ |
| | PDFS-SPGR | $0.020^*$ | 1.000 | $< .001^*$ | $< .001^*$ | $< .001^*$ | $0.018^*$ | 1.000 | $< .001^*$ | $0.004^*$ |

Note.-Pairwise comparisons were compared using Bonferroni analysis. T1 = T1-weighted, PDFS = proton density fat-saturated, SPGR = fat-suppressed 3D spoiled gradient-echo, $E_{BB}$: edge between cancellous bone and cortical bone, $E_{BC}$: edge between cortical bone and cartilage, $E_{CF}$: edge between cartilage and fat, $E_{CM}$: edge between cartilage and meniscus, $E_{BT}$: edge between cortical bone and tissue
*P-value of less than 0.05 are considered statistically significant.

**Table 4. Inter-subject variability (ISV).**

| | Sequence | Femur | | | | | Tibia | | | |
|---|---|---|---|---|---|---|---|---|---|---|
| | | $E_{BB}$ | $E_{BC}$ | $E_{CF}$ | $E_{CM}$ | $E_{BT}$ | $E_{BB}$ | $E_{BC}$ | $E_{CM}$ | $E_{BT}$ |
| Sharpness | T1 | 22.53*(42.00) | 40.58 (68.08) | 40.80 (74.64) | 24.38 (37.27) | 26.62 (34.21) | 17.64 (30.65) | 54.12 (102.7) | 23.54 (40.54) | 21.58 (35.69) |
| | PDFS | 42.97 (98.93) | 35.26 (73.09) | 34.75 (84.41) | 32.25 (40.04) | 44.09 (81.85) | 44.24 (111.46) | 39.33 (77.01) | 22.47 (29.04) | 39.69 (72.81) |
| | SPGR | 32.03 (55.92) | 30.16 (33.40) | 27.25 (54.33) | 39.02 (88.45) | 30.75 (52.31) | 30.68 (54.27) | 31.11 (33.58) | 55.59 (141.06) | 36.09 (77.69) |
| Contrast | T1 | 17.19 (25.48) | 15.78 (27.15) | 13.24 (23.75) | 15.05 (24.81) | 17.96 (33.5) | 13.69 (20.77) | 21.88 (40.90) | 15.77 (31.50) | 16.30 (34.01) |
| | PDFS | 23.54 (42.69) | 18.45 (39.66) | 12.94 (18.45) | 40.21 (108.59) | 18.83 (31.84) | 24.39 (39.3) | 34.28 (87.58) | 36.69 (92.85) | 17.04 (28.07) |
| | SPGR | 15.53 (33.45) | 18.59 (24.14) | 43.27 (74.68) | 7.71 (9.13) | 23.11 (43.06) | 15.98 (39.94) | 22.11 (23.02) | 11.45 (27.00) | 29.08 (54.96) |

Note.- T1 = T1-weighted, PDFS = proton density fat-saturated, SPGR = fat-suppressed 3D spoiled gradient-echo, $E_{BB}$: edge between cancellous bone and cortical bone, $E_{BC}$: edge between cortical bone and cartilage, $E_{CF}$: edge between cartilage and fat, $E_{CM}$: edge between cartilage and meniscus, $E_{BT}$: edge between cortical bone and tissue
*The values show the $ISV_{AVE}$ ($ISV_{MAX}$). These are average and maximum of ISV values

muscle). PDFS has a good edge between the cartilage and meniscus. Finally, SPGR showed a good edge between cortical bone and cartilage. Importantly, our experimental results showed that the edge characteristics of a structure depend on the neighboring tissue of the edges. This means that no sequence is absolutely good for all structures. Therefore, selecting a sequence suitable for the most important segmentation target will be important.

The present study devised novel metrics to evaluate edges quantitatively. However, the metrics computed, sharpness, and contrast are existing metrics known to be important for segmentation performance. However, extracting the edge of the object to be analyzed is necessary to use these metrics. There are general edge detection techniques for images [24]. However, a customized algorithm is required to extract edges from medical images according to the target structures. This is the same problem as object segmentation using medical images. To solve this problem, we manually drew the edges to measure sharpness and contrast and analyzed the edges drawn by humans. To find the best-suited MRI sequence for auto-segmentation, our proposed method can, in principle, be used for all human structures and body parts.

Our study has some limitations. First, the sequence images we used have different slice thicknesses (Table 1). In general, the sharpness and contrast of an image depend on the slice thickness [25]. Therefore, even if the same object is scanned and imaged, if it is reconstructed with different slice thicknesses, there will be a difference in noise and blurring effect. The slice thickness affects the sharpness and contrast values as well. For quantitative evaluation according to the sequence, all scan parameters other than the sequence must have the same conditions. Nevertheless, the present study aimed to present a new method. The objective was not to conduct a definitive analysis to determine which sequence is superior for a specific structure; thus, we do not consider it a critical issue.

Second, the new evaluation method proposed in this study includes metrics for measuring sharpness and contrast, but we did not compare these with the existing evaluation metrics. Concepts such as Weber-Fechner contrast [26], Michelson contrast [27], and Moon-Spencer contrast [28] that define image contrast have been proposed, but a universal definition of image contrast has not been established [29]. In addition, since our target region for contrast measurement differs from the measurement conditions suggested by existing metrics, we did not compare it with these. Our metric for contrast computes the ratio of two areas' average values, which is similar to Weber-Fechner's concept and is intuitive to compute. In the case of no reference-based metrics such as BlurMetric [30] and edge rise distance [31], they are widely used to measure image sharpness in the field of computer vision. We evaluated these metrics in our preliminary experiment but did not obtain consistent results in irregular and noisy

areas. In the case of the modulation transfer function (MTF), it has been reported that it is a suitable metric for measuring the sharpness of medical images [32]. Still, since MTF is a phantom-based sharpness measurement, it could not be applied to our study.

Finally, our method requires manual (by human) marking to define edges. In ambiguous edges, there will be differences in edge markings by different people, and this difference will lead to differences in results. This will be a limitation of our proposed technique. Therefore, selecting areas with relatively clear edges will be necessary for practical use.

Also, this study was conducted using data from healthy subjects without any specific knee pathology. In general, changes in tissue characteristics due to disease reduce the segmentation performance. This effect may be a larger variable than the difference in sequences. However, only a small amount of normal data was used since the purpose of this study was to present the methodology rather than to find the best sequence for a specific region.

## 5. Conclusions

Currently, MRI sequences vary greatly depending on equipment manufacturers, and there are hundreds of sequences, including those developed for research purposes. However, this study was conducted with only three sequences. Therefore, the results presented in this study will not be the best sequence for knee segmentation. Our purpose in this study was to present a new method to evaluate the edge sharpness and contract suitability when, for instance, determining the best-suited sequences for segmentation. In conclusion, our new method can help quantitatively compare image sharpness and contrasts from different MRI sequences for the segmentation algorithm. The proposed method can be used to aid in selecting appropriate MRI sequences for segmentation algorithms.

## Supporting information

**S1 Table. Results of the edge sharpness and contrast: Femur.**
(DOCX)

**S2 Table. Results of the edge sharpness and contrast: Tibia.**
(DOCX)

## Author Contributions

**Conceptualization:** Yunsub Jung, Michael Skipper Andersen.

**Data curation:** Yunsub Jung, Morten Bilde Simonsen.

**Formal analysis:** Yunsub Jung.

**Funding acquisition:** Michael Skipper Andersen.

**Investigation:** Yunsub Jung, Michael Skipper Andersen.

**Methodology:** Yunsub Jung, Morten Bilde Simonsen, Michael Skipper Andersen.

**Project administration:** Michael Skipper Andersen.

**Software:** Yunsub Jung.

**Supervision:** Michael Skipper Andersen.

**Validation:** Morten Bilde Simonsen.

**Visualization:** Yunsub Jung.

**Writing – original draft:** Yunsub Jung, Morten Bilde Simonsen.

**Writing – review & editing:** Michael Skipper Andersen.

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
