## [Decision Letter · Decision Letter 0]

12 Aug 2024

PONE-D-24-07156A Method for Comparing MRI Sequences of the Knee for Segmentation Based on Morphological FeaturesPLOS ONE

Dear Dr. Andersen,

Thank you for submitting your manuscript to PLOS ONE. After careful consideration, we feel that it has merit but does not fully meet PLOS ONE’s publication criteria as it currently stands. Therefore, we invite you to submit a revised version of the manuscript that addresses the points raised during the review process.

We look forward to receiving your revised manuscript.

Kind regards,

Emil George Haritinian, M.D, Ph.D.

Academic Editor

PLOS ONE

“This work supported by the Novo Nordisk Foundation (grant no. NNF21OC0065373).”

Additional Editor Comments:

I would like to congratulate the authors on their engaging study. However, as the reviewers have noted, there are some minor corrections needed, along with several areas that could be further enhanced. I look forward to reviewing the revised version of the manuscript.

Reviewers' comments:

Reviewer's Responses to Questions

**Comments to the Author**

1. Is the manuscript technically sound, and do the data support the conclusions?

Reviewer #1: Partly

Reviewer #2: Yes

Reviewer #3: Partly

2. Has the statistical analysis been performed appropriately and rigorously? 

Reviewer #1: N/A

Reviewer #2: I Don't Know

Reviewer #3: Yes

3. Have the authors made all data underlying the findings in their manuscript fully available?

Reviewer #1: No

Reviewer #2: Yes

Reviewer #3: Yes

4. Is the manuscript presented in an intelligible fashion and written in standard English?

Reviewer #1: Yes

Reviewer #2: Yes

Reviewer #3: Yes

5. Review Comments to the Author

Reviewer #1: Remove all typos from the text.

The abstract should be improved (conclusions in the abstract!)

The Nomenclature part should be added for all abbreviations and symbols used in the text.

What is the limitations of the used technique?

Add references to the equations used !

Add more details to the proposed image processing algorithm.

Enrich the introduction part with recent articles on Otsu’s thresholding technique for MRI image brain tumor segmentation.

The discussion part should be improved.

Write the conclusion part in the form of important points.

Reviewer #2: The authors of the study aim to analyze a new method for automatic/semi-automatic segmentation of structures in magnetic resonance imaging (in this case, the knee, but with the possibility of extending to other regions or structures). Three types of sequences were chosen: T1, PD, and SPGR, and five kinds of edges between structures (cortical bone-cancelous bone, cortical bone-cartilage, cartilage-fat, cartilage-meniscus, cortical bone-tissue).

The authors explain why two features - edge sharpness and contrast between two different types of tissue—are so important for segmentation methods (semi-automatic or using machine learning). The authors do not intend to decide which is the best sequence for diagnosis or segmentation of a specific structure, but to provide a tool to decide, depending on the structure we need to segment, which sequence would be the best, as clearly outlined in the discussion chapter.

The authors explain in detail the image acquisition technique on healthy volunteers using standard practice MRI sequences. The usual parameters of these sequences are explained, specifically in what types of pathologies they can be useful.

Using elaborate mathematical methods, after manually tracing the edges, the authors calculate contrast and edge sharpness indices to decide which sequence is more useful for which type of segmentation.

The authors highlight the difficulty of tracing the edges of structures in the human body, a process different from tracing edges between other types of materials, and explain why manual (human) tracing was chosen, followed by computerized analysis of this human tracing.

The authors acknowledge limitations of the study, such as the different slice thickness between the three types of sequences, which may influence the analysis. However, the study aims to establish a method of analysis, not to provide the answer to which sequence is the best.

One of the important conclusions is that none of the sequences is perfect for any type of structure we want to segment.

It should be noted that the study aims to explain the use of a new analysis method to serve as a decision tool for choosing the best sequence concerning the structure we want to segment (cartilage, cortical bone, spongy bone, etc.).

The study has a clear, well-explained methodology, and the conclusions are well-defined. For radiologists and imaging researchers this study provides a tool that might prove to be very useful.

Areas for improvement:

Lines 220-226 suggestion: The information is quite complex; it might be useful to synthesize it in a table: for edge sharpness and for contrast - with each sequence and the conclusion or which sequence was the best for which type of edge detection.

Although it may not be important given the purpose of using a new segmentation method, the fact that healthy subjects without particular knee pathologies were used could be one of the study's limitations and maybe mention this in limitations.

Line 148 proton density-weighted (PD) - there is an imaging inconsistency: images shown as an example - line 376 - the image annotated as PD is in fact “PD with fat saturation”. This propagates throughout the article and deserves further clarification: is the sequence tested actually PD or PD with fat saturation?

Lines 152-154 - there might be a slight misunderstanding - on SPGR sequences, bone marrow is difficult to interpret, and it certainly does not come into direct contact with the cartilage (the cartilage is separated from the bone marrow by the cortical bone).

Lines 121 to 123 are a bit confusing, maybe that phrase can be reformulated.

Line 124 - “materials” refers in the human body as “tissue,” and maybe the use of “tissue” or “tissular” could be a

more appropriate term when referring to human MRI images.

Reviewer #3: The sequences are not fully named in the text and in the fig. 1. Thus: the first 2 sequences are annotated with the weighting type of (T1 and PD), and the third one, with the type of sequence (SPGR). Both annotations must be made (respectively T1 FSE/SE, PD FSE/SE and 3D (probably) T1 FSPGR.

In addition, both the PD sequence and the FSPGR are fat saturation acquisitions, which should also be noted in the text and on the image.

Was the data tested for normality? Was ANOVA the most appropriate test or was a nonparametric test needed?

The big difference in the acquisition parameters of the sequences used for comparison and the small number of patients included in the study are the major limitations of this study. From these two points of view, I think that the study could be improved.

6. PLOS authors have the option to publish the peer review history of their article (what does this mean?). If published, this will include your full peer review and any attached files.

Reviewer #1: **Yes: **Prof. Dr. Fateh Mebarek-Oudina

Reviewer #2: **Yes: **Sorin Ghiea

Reviewer #3: **Yes: **Emi Marinela Preda

---

## [Author Response · Author response to Decision Letter 0]

22 Aug 2024

This update is for revision (22.Aug.2024)

---

## [Decision Letter · Decision Letter 1]

20 Sep 2024

A Method for Comparing MRI Sequences of the Knee for Segmentation Based on Morphological Features

PONE-D-24-07156R1

Dear Dr. Andersen,

We’re pleased to inform you that your manuscript has been judged scientifically suitable for publication and will be formally accepted for publication once it meets all outstanding technical requirements.

Kind regards,

Mylène P. Jansen, PhD

Academic Editor

PLOS ONE

Additional Editor Comments (optional):

Thank you for revising your manuscript as requested by the reviewers. As they were generally satisfied with your changes and motivation, the paper can be accepted for publication in PLOS ONE.

Reviewers' comments:

Reviewer's Responses to Questions

**Comments to the Author**

1. If the authors have adequately addressed your comments raised in a previous round of review and you feel that this manuscript is now acceptable for publication, you may indicate that here to bypass the “Comments to the Author” section, enter your conflict of interest statement in the “Confidential to Editor” section, and submit your "Accept" recommendation.

Reviewer #1: (No Response)

Reviewer #2: All comments have been addressed

2. Is the manuscript technically sound, and do the data support the conclusions?

Reviewer #1: Partly

Reviewer #2: Yes

3. Has the statistical analysis been performed appropriately and rigorously? 

Reviewer #1: N/A

Reviewer #2: I Don't Know

4. Have the authors made all data underlying the findings in their manuscript fully available?

Reviewer #1: Yes

Reviewer #2: Yes

5. Is the manuscript presented in an intelligible fashion and written in standard English?

Reviewer #1: Yes

Reviewer #2: Yes

6. Review Comments to the Author

Reviewer #1: General Comments

The manuscript presents valuable research; however, several areas require attention to enhance clarity and comprehensiveness. Below are specific comments and suggestions for improvement.

Specific Comments

Typographical Errors:

The author should thoroughly review the manuscript for typographical errors throughout the text. A careful proofreading will help ensure that the presentation is polished and professional.

Nomenclature:

It is essential to verify that all abbreviations and symbols used in the text are clearly defined in the Nomenclature section. This will aid readers in understanding the terminology and enhance the manuscript's accessibility.

Limitations of the Technique:

The limitations of the techniques employed in the study should be explicitly discussed. A clear presentation of these limitations will provide a balanced view of the research findings and help readers understand the context and applicability of the results.

Introduction Enrichment:

The introduction should be enriched with recent literature, particularly focusing on advancements such as Otsu’s thresholding technique for MRI image brain tumor segmentation. Including this information will provide a more comprehensive background and highlight the relevance of the current study within the broader research landscape.

Discussion Adequacy:

The discussion section needs to be more adequate and thorough. It should critically analyze the results, compare them with existing literature, and explore the implications of the findings in greater depth. This will strengthen the manuscript and provide readers with a clearer understanding of the significance of the research.

Conclusion

Addressing these comments will significantly improve the quality of the manuscript. A thorough revision focusing on typographical errors, nomenclature clarity, limitations, literature enrichment, and a more robust discussion will enhance the overall impact of the research presented.

Reviewer #2: thank your for the changes that you made after my initial review. I wish you a all the best in your future endeavors

7. PLOS authors have the option to publish the peer review history of their article (what does this mean?). If published, this will include your full peer review and any attached files.

Reviewer #1: No

Reviewer #2: **Yes: **Dr. Sorin Ghiea

---

## [Editor Report · Acceptance letter]

26 Sep 2024

PONE-D-24-07156R1 

PLOS ONE

Dear Dr. Andersen, 

I'm pleased to inform you that your manuscript has been deemed suitable for publication in PLOS ONE. Congratulations! Your manuscript is now being handed over to our production team.

Kind regards, 

on behalf of

Dr. Mylène P. Jansen 

Academic Editor

PLOS ONE